# Beyond Cancer: Regulation and Function of PD-L1 in Health and Immune-Related Diseases

**DOI:** 10.3390/ijms23158599

**Published:** 2022-08-02

**Authors:** Amke C. Beenen, Tatjana Sauerer, Niels Schaft, Jan Dörrie

**Affiliations:** 1Department of Dermatology, Universitätsklinikum Erlangen, Friedrich-Alexander-Universität Erlangen-Nürnberg, Hartmannstraße 14, 91052 Erlangen, Germany; amke.beenen@fau.de (A.C.B.); tatjana.sauerer@uk-erlangen.de (T.S.); niels.schaft@uk-erlangen.de (N.S.); 2Comprehensive Cancer Center Erlangen European Metropolitan Area of Nuremberg (CCC ER-EMN), Östliche Stadtmauerstraße 30, 91054 Erlangen, Germany; 3Deutsches Zentrum Immuntherapie (DZI), Ulmenweg 18, 91054 Erlangen, Germany

**Keywords:** PD-L1, non-cancerous tissues, immune checkpoint, PD-1–PD-L1 axis

## Abstract

Programmed Cell Death 1 Ligand 1 (PD-L1, CD274, B7-H1) is a transmembrane protein which is strongly involved in immune modulation, serving as checkpoint regulator. Interaction with its receptor, Programmed Cell Death Protein 1 (PD-1), induces an immune-suppressive signal, which modulates the activity of T cells and other effector cells. This mediates peripheral tolerance and contributes to tumor immune escape. PD-L1 became famous due to its deployment in cancer therapy, where blockage of PD-L1 with the help of therapeutic antagonistic antibodies achieved impressive clinical responses by reactivating effector cell functions against tumor cells. Therefore, in the past, the focus has been placed on PD-L1 expression and its function in various malignant cells, whereas its role in healthy tissue and diseases apart from cancer remained largely neglected. In this review, we summarize the function of PD-L1 in non-cancerous cells, outlining its discovery and origin, as well as its involvement in different cellular and immune-related processes. We provide an overview of transcriptional and translational regulation, and expression patterns of PD-L1 in different cells and organs, and illuminate the involvement of PD-L1 in different autoimmune diseases as well as in the context of transplantation and pregnancy.

## 1. Introduction

The classical T-cell immune response involves the presence of an antigenic protein, which is processed and presented by antigen presenting cells (APCs). For the downstream activation and clonal expansion of antigen-specific T cells, different signals have to be delivered when the APC comes into contact with the naïve T cells. This ensures a specific and regulated activation of the T-cell clones needed to target the current pathogenic antigen [1].

The first required signal is the matching interaction of the APC’s major histocompatibility complex (MHC) and the T-cell receptor (TCR) involving the MHC-presented antigen and different co-receptors, in order to ensure an interaction leading to downstream signaling and the activation of the T cell. In addition, a co-stimulatory signal is required, which, as opposed to the first signal, is antigen-independent and delivered by different proteins expressed on the APC, which come in contact with their matching ligands on T cells [2]. A combination of these signals leads to the activation of several cellular pathways, including the Mitogen-Activated Protein Kinase (MAPK) and the Nuclear Factor kappa-light-chain-enhancer of activated B cells (NFκB) pathways, which then trigger the expression of various cytokines. These cytokines compose the third signal, which transmits through the mTOR and PI3K pathway and is crucial for the activation of the cells’ proliferation. Only a combination of all signals leads to a lasting T-cell activation and clonal expansion [3].

Focus is placed here on the second signal, where co-stimulatory molecules on the APC are interacting with the respective receptors on the T cell. A variety of these co-stimulatory interactions exist, notably, the B7-1 and B7-2:CD28 interaction. This is a stimulatory interaction leading to productive T-cell activation and survival, given the presence of the other signals [4]. However, CD28 is not the only binding partner of B7-1 and B7-2—activated T cells upregulate the inhibitory receptor CTLA4 (Cytotoxic T-Lymphocyte-Associated protein 4), which also acts as an immune checkpoint to prevent an overshooting of the T-cell-mediated immune response [5]. PD-L1 is another member of the B7 family, but exerts suppressive functions. When binding to its receptor, PD-1, on the T cell, it causes an inhibition of T-cell effector functions and proliferation, locally leveling down the immune system. This is one of several important mechanisms for the maintenance of peripheral tolerance and prevention of autoimmunity. Peripheral tolerance represents a second line of defense to avoid autoimmunity caused by immune cells that escaped the selection-mediated central tolerance mechanisms [6].

## 2. PD-L1 and Its Interaction Partners

The engagement of PD-L1 with its receptor PD-1, as depicted in Figure 1, suppresses T-cell proliferation and cytokine production, as shown by Freeman et al. [7] using purified CD4-positive T cells. When activated with high concentrations of anti-CD3 antibodies, which provide a TCR stimulatory signal, initiating proliferation, the subsequent PD-1–PD-L1 interaction reduced that proliferation. Cytokine levels secreted by T cells dropped significantly for IFNγ and IL-10, and even below a detectable threshold for IL-2 [7]. The addition of an agonistic anti-CD28 antibody at varying concentrations, providing co-stimulation, revealed that the inhibitory effect on proliferation by PD-L1 depends on the strength of the TCR and CD28 signals, as high concentrations lowered the grade of suppression by PD-L1. When, however, T cells were provided with optimal anti-CD3 stimulation, a significant repressive effect of PD-L1 was only observed in the complete absence of co-stimulation [7]. This shows that upon strong TCR stimulation, cytokine production is decreased via PD-L1, but T-cell proliferation is not inhibited. The mechanism that suppresses proliferation has been shown to be cell cycle arrest in the G0/G1 phase by PD-L1, rather than apoptosis [8].

PD-L1 can also directly bind to CD80 with a higher affinity for CD80 than CD28, which leads to the dissolution of the CD80 heterodimer, lowering the spectrum of potential CD80 binding partners [9]. The CD80–PD-L1 interaction can lead to a partial inhibition of the PD-L1–PD-1 as well as CTLA4-mediated CD80 depletion through the prevention of CD80 homodimer formation, while keeping CD80–CD28 signaling intact [10] (Figure 1). Antibodies blocking the PD-L1–CD80 interaction were shown to alleviate autoimmunity by restoring PD-L1–PD1 interaction [11].

These findings highlight the classical function of PD-L1 of being capable of leveling down immune cell responses in various cell types, reducing their proliferation and effective capabilities.

## 3. Perpetual Presence of Antigen Leads to Loss of T-Cell Function via PD-1–PD-L1 Interaction

After pathogen clearance, the specific lymphocyte pool dies by apoptosis, with a small fraction developing into memory cells capable of initiating effector functions quickly upon secondary antigen contact. If, in contrast, the immune system is unable to remove the antigen, the phenotype of exhausted lymphocytes emerges. Target cells, who upregulate their PD-L1 expression under chronic infection, trigger PD-1 on the T cells. This interaction leads to the suppression of T-cell expansion and inhibits their function. Loss of function was reported to be timely and hierarchically regulated, with the expression of TNF persisting longer than IL-2 [12]. Classically, this occurs in chronic infections with viruses such as HIV or HCV, or in malaria [13].

Blockage of PD-1–PD-L1 interaction in these infections led to a restoration of T-cell functions, including proliferation, and a reduction in viral load selectively in already exhausted T cells. This is of therapeutic relevance since CD8 T cells specific for HIV or HCV express higher PD-1 levels. Noteworthy is that even after functional restoration through PD-L1 blockage, T cells still expressed elevated PD-1 levels, suggesting that especially prolonged PD-1–PD-L1 interaction is leading to the exhausted phenotype [14,15]. Pathogens have been shown to hijack this mechanism in order to facilitate immune evasion, inducing PD-L1 expression on immune cells. The parasite *Schistosoma mansoni* is capable of inducing T-cell anergy through the selective upregulation of PD-L1 surface expression on macrophages, suppressing pathogen clearance [16].

## 4. Discovery and Classification of PD-L1

The PD-L1 or B7-H1 is a member of the B7 family. This family consists of seven cell-surface proteins, which are structurally related and are capable of binding to lymphocyte receptors, enabling immune response modulation [17]. Depending on the protein, members of the B7 family are able to deploy stimulatory as well as attenuating signals, acting mostly on T cells via their respective receptors [17,18].

PD-L1 was independently discovered in the late 20th century by two individual working groups. Dong et al. in the lab of Lieping Chen identified a new homologue to B7-1 and B7-2, which did not act as ligand for CD28 and ICOS. They termed the new protein B7-H1 and reported a co-stimulatory function [19]. Subsequently, Gordon Freeman et al., in the lab of Tasuko Honjo, were able to identify B7-H1 as the ligand for PD-1 in both mice and humans by using a PD-1-Ig fusion protein. They renamed the protein PD-L1 and demonstrated its inhibitory function on TCR-induced T-cell proliferation and cytokine release [7].

## 5. Genetic Location and Structure of PD-L1 and Its Interaction with PD-1

The gene encoding PD-L1, *CD274*, in humans is comprised of seven exons, which are located on the 9th chromosome at 9p24.1 between 5.45 Mbp and 5.47 Mbp. The first exon is non-coding, whereas the following exon contains the start codon and the signal peptide. Exon 3 and 4 encode the immunoglobulin (Ig)-V-like and Ig-C-like domains. Exon 5 and 6 contain the transmembrane domain and the intracellular domain with some parts also located in the last exon, which otherwise contains the 3’UTR [20].

The transcript can be differentially spliced, with the 4.2 kbp transcript being the major one, along with a 7.2 kbp transcript and one truncated form missing the Ig-V-like domain [7,19,21]. Full-length PD-L1 is a 31 kDa protein, consisting of 290 amino acids [20,22]. The extracellular part was shown to have a homology of about 20% amino acid identity with B7-1 [19,23].

Structurally, PD-L1 is an Ig-like transmembrane glycoprotein receptor which is located at the plasma membrane surface. Functionally, there are two domains present which are linker-joined and fold in an Ig-like manner [24]. The *N*-terminal domain of PD-L1, responsible for PD-1 interaction, is characterized by the Ig-V-type topology while also being equipped with an Ig-C-type extracellular domain whose function remains unidentified [23,24,25]. The intracellular part was found to be short, containing no known motifs capable of signal transduction, although there are some hints that this part is important for the transduction of survival signals [20,23,26].

The interaction of PD-L1 and PD-1 was visualized using crystal structures by Zak et al. [25] at a resolution of 2.45 Å for up to 99.98% completeness. The dimerization of PD-L1, which has been theorized before, has been disproved, and a PD-1–PD-L1 complex as a 1:1 combination with an Ig-V-domain antibody-like interaction resemblance has been accepted. Both binding partners have been shown to be located orthogonally to each other, leaving the complementarity-determining region (CDR) loops out of the interaction, thereby forming a structure, which was described as resembling the antigen-binding site within classical antibodies. The total interaction surface is 1970 Å² and involves both polar as well as hydrophobic interactions [25].

## 6. Pathway of PD-L1 Expression

Classical expression of the PD-L1-encoding gene *CD274* is known to be initiated mainly by activation of the Janus Kinase (JAK)/Signal Transducer and Activators of Transcription (STAT) pathway, which is triggered by inflammation-modulating cytokines. These modulators include especially interferons (IFNs) and are released by activated immune cells [27].

The signal stemming from IFN–IFN-receptor interaction is transferred intracellularly through Tyrosine Kinase (TYK) 2/JAK1 and JAK1/JAK2 and leads to the phosphorylation of different STAT members and the subsequent formation of several complexes. Different STAT factors play a role, with STAT1, STAT2, and STAT3 being mostly involved. Additional regulation is mediated by Interferon Regulatory Factors (IRF) 1 and IRF9. All these transcription factors were found to be upregulated after IFN stimulation in a positive feedback loop [28]. The most notable of the complexes formed are the IFN-Stimulated Gene Factor 3 (ISGF3) complex, consisting of STAT1/STAT2/IRF9, and the Gamma-Activating Factor (GAF) complex, consisting of a phosphorylated STAT1 dimer. Both complexes are then able to translocate to the nucleus and act as transcription factors on a Gamma-Activated Sequence (GAS) on IRF1, whereas the ISGF3 complex is also capable on inducing Interferon Stimulated Response Element (ISRE) activity. This leads to the induction of several interferon-induced genes, whereas the GAS interaction then initiates IRF1 expression, which is capable of binding to the PD-L1 promotor, enabling its transcription [28,29,30]. The importance of the IRF1 binding sites has been shown by Lee et al. [31], highlighting their importance in constitutive as well as inducible expression.

Additional pathways are of relevance for PD-L1 expression, with the MEK/ERK pathway being crucial in monocyte-derived dendritic cells (moDCs) [32], and the PI3K-AKT and MAPK-ERK pathway increasing PD-L1 expression. The capability of PD-1 signaling to downregulate these two pathways could generate a negative feedback loop in cells expressing both the ligand and the receptor [33,34].

The inhibition of the PI3K pathway reduced the increase in PD-L1 expression in cytokine-stimulated moDCs, whereas the depletion of p38, a member of the MAP kinases, did not exert a comparable effect. In myeloid DCs (mDCs), p38 was shown to have a more pronounced relevance, as both ERK and p38 activation were required for the cytokine-induced PD-L1 increase, whereas in freshly isolated plasmacytoid DCs (pDCs), the induced expression was more reliant on the p38 pathway [32].

In short, the JAK/STAT pathway initiated through IFN is the main and classical pathway to initiate PD-L1 expression, with other pathways being of importance under certain conditions.

## 7. Modulation of PD-L1 Expression by Different Cytokines and Transcription Factors

### 7.1. IFN Induction of PD-L1

IFNγ, the solitary type II IFN representative, is one of the most prominent examples leading to a strong expression of PD-L1. Type I IFN exposure was shown to lead to lower PD-L1 expression due to lower IRF1 promotor binding when compared to type II IFN treatment [28]. Nevertheless, Bazhin et al. [35] found that IFNα stimulation in DCs increased the expression of PD-L1 via STAT3 and p38 activation, complemented by an upregulation of IL-6 and downregulation of IL-12, respectively. This was shown to result in a lowered IFNγ production induction capacity in T cells by stimulated DCs [35]. In B and T cells, as well as endo- and epithelial cells, type I and II IFNs together with TNF were shown to induce PD-L1 expression [36].

Site-directed mutagenesis of the PD-L1 promotor region in a reporter system showed that selective deletion of a putative IRF1 site significantly decreased PD-L1 expression, highlighting its crucial role in PD-L1 transcription regulation [28]. In contrast, deletion of a putative STAT1/STAT3 binding site showed an increase in reporter expression, hinting towards a potential repressor also capable of binding, or other elements such as silencers, all leading to a negative effect on PD-L1 expression in the presence of a functional binding site [28].

The inhibition of type I and II IFN signaling-related genes was found to affect PD-L1 expression in different severities. The limiting factors found by Garcia-Diaz et al. [28] as tested through depletion were identified as JAK1, JAK2, and TYK2, as crucial translators of the IFN signal upstream, and IRF1 and IRF9 of that downstream. STAT1, STAT2, and STAT3 only had a minor influence on the resulting expression level, probably due to a redundancy of the different STAT factors. Previous works on JAK1 and JAK2 have established the importance of their functions in PD-L1 expression, as knockout led to a loss in mRNA upregulation even when stimulated with IFNγ [37].

The prominent role of IFNγ in the induction of PD-L1 expression hints towards its role in mediating Th1-T-cell functions. Differential studies comparing Th1- and Th2-mediated protein induction found that Th1-T cells, when confronted with APC-presented peptides, are significantly more capable of inducing PD-L1 expression, which was found to be STAT1 modulated, matching with the classical pathway known for PD-L1 expression [38]. The upregulation of PD-L1 on inflammatory macrophages was found to be primarily dependent on type I cytokines compared to type II cytokines, with the latter only capable of inducing minor PD-L1 expression, which is why upregulation through type II cytokines has been classified as alternative stimulation [38].

### 7.2. Other Common Gamma Chain Interleukines

Besides IFN-signaling, those interleukins that signal via the cytokine receptor common subunit gamma (IL2RG), which is also capable of activating STAT1 and STAT3 [39,40] (i.e., IL-2, IL-7, IL-15, and IL-21), can induce PD-L1 expression [41]. Whereas most of these are capable of inducing PD-L1 expression on T cells, IL-21 was shown to not exert this effect. It was, however, capable of stimulating protein expression in B cells [36]. The immunosuppressive cytokine IL-27 was also shown to induce PD-L1 expression via STAT1 activation [42]. TNF, being another crucial pro-inflammatory cytokine with a distinct signaling pathway, was shown to synergize with IFN in PD-L1 upregulation by increasing the expression of the IFN receptors [43]. IL-17 was capable of further boosting this induction via a nitric-oxide-dependent mechanism [44].

Additionally, IL-10 was found to be able to stimulate PD-L1 expression in monocytes [45] in a STAT3-dependent manner [46]. It can, however, have contrary effects based on the cell type, as IL-10 led to a decrease in PD-L1 expression in cancer cell studies via an inhibition of the MAPK pathway, potentially due to the higher IL-10 levels in these tissues [47].

### 7.3. Activators of NFκB

Lipopolysaccharide (LPS), a Pathogen Associated Molecular Pattern (PAMP) that signals via the Toll-Like Receptor 4 (TLR4) and triggers NFκB, was shown to lead to the association of NFκB to the PD-L1 promotor and its subsequent expression in colorectal cancer cells, extending the evidence found from primary cells [48]. Studies in human monocytes revealed the presence of an NFκB binding site inside the *CD274* gene crucial for responding upon the presence of PAMPs such as LPS. NFκB inhibitors significantly dampened upregulation, thus showing the relevance of the NFκB pathway for PD-L1-regulation by PAMP activation [49]. IL-18, a cytokine, that also triggers the NFκB pathway, was also shown to upregulate PD-L1 expression in regulatory B cells (Breg) [50].

### 7.4. Negative Regulators of PD-L1 Induction

Numerous inhibitory factors have been identified, capable of modulating several steps along the signaling cascade. Upstream factors such as the tyrosine-protein phosphatase non-receptor type (PTPN)11, PTPN6, and Protein Tyrosine Phosphatase Receptor type C (PTPRC) are inhibiting JAK1 phosphorylation, whereas Suppressor Of Cytokine Signaling (SOCS)1, SOCS2, and SOCS3 are acting on the STAT factors further downstream. The complexes formed by phosphorylated STAT factors and the accompanying interferon regulatory factors can be inhibited by Protein Inhibitor of Activated STAT1 (PIAS)1 and PIAS3 [28].

These results show the diversity in PD-L1 expression regulators and differences between various cell types, with IFNγ being the major inducer. An overview of the influence of various cytokines and transcription factors is given in Table 1 and their interaction is further visualized in Figure 2.

## 8. Post-Transcriptional Regulation and Modulation

As PD-L1 is capable of reducing the immune response, several mechanisms have evolved to regulate protein expression and half-life time.

### 8.1. Regulation on mRNA Level

RNA interference (RNAi) is a mechanism involving microRNAs (miRNAs) which form duplexes with mRNAs due to perfect or imperfect complementarity, resulting in mRNA degradation or the modulation of translation [54]. Several miRNA play a role in PD-L1 regulation [55]. Biliary epithelial cells were found to express miRNA-513, targeting the PD-L1 3’UTR, thus being able to repress PD-L1 translation. This miRNA was downregulated upon IFNγ exposure, thus amplifying the IFN-induced production of PD-L1 [56].

Studies on differentially regulated miRNAs in dermal lymphatic endothelial cells and fibroblasts unveiled an upregulation of miRNA-155 induced by TNF and enhanced by IFNγ. miRNA-155 was also shown to act on the 3’UTR at two appropriate binding sites. An overexpression of miRNA-155 led to an increase in PD-L1 mRNA levels in endothelial cells but not in fibroblasts. This did not transfer to the protein level, as suppression of the PD-L1 protein expression was observed, and miRNA-155 depletion achieved the contrary effect. Interestingly, evidence towards a regulatory miRNA network was found, as induction of the PD-L1 expression through IFN stimulation simultaneously activated repressing components. This is a mechanism potentially ensuring the intracellular balance to prevent ongoing immune suppression [57].

Other miRNAs exerting different functions on PD-L1 expression have been identified, with their role, however, mostly being pronounced in cancer tissue. These include miRNA-34 and miRNA-873, capable of inhibiting PD-L1 expression, miRNA-33a and miRNA-21, which suppress PD-L1 expression, and miRNA-873, which promote it [55]. Efforts were taken to understand the mechanism of how tumor cells regulate PD-L1 to potentially counteract it and prolong patient survival, and the miRNAs-152, -200b, -142-5p, and -197 have been identified to be of importance in different cancer types. However, their role under physiological conditions and in healthy tissue still remains unclear [58,59,60].

A different mechanism regulating mRNA stability is AU-rich element (ARE)-mediated decay. Epidermal growth factor receptor (EGFR) activation is capable of stabilizing PD-L1 mRNA by preventing this, resulting in an increase in PD-L1 protein levels [61]. A similar mechanism was described for RAS signaling [62]. Both pathways were downregulating ARE-binding proteins, thus preventing RNA decay.

### 8.2. Regulation on Protein Level

PD-L1 regulation of the protein level is mostly achieved through the poly-ubiquitination and subsequent degradation by the proteasome. The NFκB pathway component p65 induces the CSN5 (COP9 signalosome 5), which is capable of inhibiting PD-L1 ubiquitination and therefore prolonging protein presence [63]. Research regarding the involvement of the NFκB pathway, however, has brought up contradictory findings. Pathway inhibition in DCs was shown to lead to tolerogenic DCs, expressing higher PD-L1 levels, whereas other studies showed that the CSN5 was required to enhance PD-L1 protein stability, which would lead to a longer persisting protein expression [63,64].

The cytokine TNF, capable of PD-L1 mRNA induction, but only when combined with other cytokines as discovered by Wang et al. [44], was subsequently linked to protein stabilization. A combination of a protease inhibitor and TNF significantly increased protein expression levels while not affecting mRNA expression as much, which is in contrast to IFNγ treatment, which upregulates PD-L1 on the mRNA level. A detailed analysis of the pathway revealed the involvement of p65 shown through IκB kinase β (IKKβ) inhibitors and p65 overexpression and knockout. It was found that TNF stimulation led to the translocation of p65 into the nucleus, where it was stabilized, and activated the transcription of CSN5, which was functionally required for the PD-L1 regulation. This showed the translational level of regulation through TNF treatment, as transcriptional levels remained unchanged [63].

The glycogen synthase kinase 3β (GSK3β) was shown to induce the proteasomal degradation of unglycosylated PD-L1 through binding and phosphorylation. This can be counteracted by an active EGFR, which is capable of stabilizing the PD-L1 transcript via GSK3β inactivation. *N*-Glycosylation of PD-L1, a mechanism which mitigates GSK3β effects and stabilizes PD-L1, was, however, only found in cancer cells until now [65].

## 9. Distribution of PD-L1 on Different Cell Types and Organs

PD-L1 expression was mainly addressed in the context of tumor immunotherapy (reviewed in [66]). Due to its crucial role in immune regulation, it is of high interest which cells and tissues show expression of this key immune molecule in healthy tissue and diseases apart from cancer.

### 9.1. PD-L1 Expression in Different Tissues

Early studies performed shortly after the initial discovery of PD-L1 aimed to screen the human body for the presence of PD-L1. Various non-lymphoid tissues were shown to have PD-L1 mRNA present, as investigated using Northern Blots [19]. These tissues included the heart, lung, placenta, and skeletal muscle, with abundant PD-L1 expression. Lower PD-L1 levels were detected in the kidney, liver, spleen, and thymus. The brain, colon, and small intestinal tissue were found to be free of PD-L1 mRNA [19]. Subsequent analysis, however, uncovered that even in tissues expressing measurable PD-L1 mRNA levels, functional protein could not be detected.

Dong et al. [67] examined tissue obtained from the breast, colon, kidney, lung, pancreas, skeletal muscle, and uterus using monoclonal antibodies. However, no immunoreactivity was measurable in these healthy tissues. Interestingly, even in tumor tissue expressing PD-L1, the adjacent healthy tissue was not found to express PD-L1, indicating a focal and concentrated tissue expression [67].

Although not detectable at baseline conditions, PD-L1 was shown to be inducible in different organs such as the heart and placenta, where, especially in the latter, it is relevant for the maintenance of fetal tolerance during pregnancy [68]. However, information about the PD-L1 expression of the placenta strongly varies: referring to the human protein atlas, medium levels of PD-L1 on the mRNA level and high expression of PD-L1 on the protein level were detected. The expression and function of PD-L1 in the prevention of fetal rejection is discussed below in detail.

The human protein atlas provides a comprehensive picture of PD-L1 expression for many organs on both the mRNA and protein levels. However, it does not completely match the expression data described in the literature: the respiratory system showed high PD-L1 expression on both the mRNA and protein levels, whereas the gastrointestinal tract, the pancreas, and the kidney demonstrated a low, medium, and medium expression of PD-L1, respectively.

One of the rare exceptions of organs where PD-L1 can be found constitutively expressed without any external triggers is the cornea and retinal pigmented epithelium, where it has been shown to deflect activated T cells from exerting their function in the immune-privileged eye, as PD-1 expressing CD4- and CD8-positive T cells underwent apoptosis after infiltration [27]. In contrast, the human protein atlas documents only a slight expression of PD-L1 mRNA in the eye and does not provide any data on PD-L1 protein levels. An overview of PD-L1 expression on the mRNA and protein levels in different tissues based on the human protein atlas can be seen in Figure 3. The results show that although PD-L1 mRNA is present in many organs and tissues, only the minority of these tissues actually expresses functional protein.

### 9.2. Various Cells Are Capable of PD-L1 Expression

Even though whole organs have been found to be largely PD-L1 negative, various hematopoietic and non-hematopoietic cell populations express PD-L1 under healthy, baseline, and activated conditions, most notably the fraction of immune cells. Again, a difference between mRNA and protein expression patterns was observed.

Both CD4- and CD8-positive T cells are PD-L1 positive at baseline, although to a limited extent, but the expression increases upon activation, applying also to CD4-positive regulatory T cells (Tregs). Exhausted T cells, however, although showing a similar baseline expression, decrease their PD-L1 expression upon stimulation [36]. A subtype of Tregs capable of PD-L1 expression was identified in multiple sclerosis research, functionally involved in the prevention of disease progression [71,72].

Freshly isolated B cells, similar to their T-cell counterpart, were shown to express only low levels of PD-L1, which increased only slightly upon LPS treatment and more noticeably upon IL-21 stimulation or BCR activation [27,73]. Some gut mucosal IgA plasma cells in the small intestine lamina propria, however, express high levels of PD-L1 [74] (see below). Regulatory B cells (Bregs) are a subset of regulatory cells working towards the maintenance of immune tolerance through the secretion of tolerogenic cytokines such as IL-10. They have been shown to express PD-L1 at baseline, but expression increased upon IL-18 stimulation. The interleukin treatment was also promoting the phosphorylation of p65, which is important in the translational regulation of PD-L1 [50]. Khan *et al.* also described a PD-L1-high population of B cells in human blood, which they characterized mainly as Bregs [75]. Natural killer (NK) cells from healthy donors expressed limited amounts of PD-L1. However, levels increased after incubation with tumor cells, with IL-18 shown to be critical in this process [71].

A notable fraction of CD14-positive monocytes are also already PD-L1 positive at baseline, while expression is enhanced even further upon IFNγ stimulation [19]. Both myeloid DCs (mDCs), and plasmacytoid DCs (pDCs) are inducible to express PD-L1 at high levels, with mDCs expressing slightly higher levels [32]. An assessment of subsets of dendritic cells revealed that the expression of PD-L1 in monocyte-derived DCs (moDCs) and total blood DCs was significantly upregulated in both cell types upon stimulation with a cytokine cocktail of TNF, IL-6, IL-1β, and PGE_2_, and also with LPS and poly I:C. DCs freshly isolated from healthy donor blood only had limited PD-L1 baseline expression. The separation of mDC and pDC revealed that the latter only responded to CpG and weakly to the cytokine cocktail, while mDC responded to poly I:C, cocktail, and LPS, but not to CpG, indicating a broader expression ability in mDCs compared to pDCs [32].

Monocytes, not expressing PD-L1 mRNA in steady state, were capable of upregulation upon IFNγ stimulation [7]. Another myeloid cell type expressing detectable levels of PD-L1 are macrophages located in the liver, lung, and tonsil. Pulmonary macrophages expressing PD-L1 have been found during the attack of PD-L1-positive tumors; however, not all macrophages in tumor proximity were detected to be positive [67].

Iwai et al. showed constitutive expression of the PD-L1 protein on non-parenchymal liver cells, which they could map to sinusoidal endothelial cells and Kupffer cells (i.e., liver-resident macrophages) [76]. Vascular and microvascular endothelium cells were found to have low PD-L1 expression but to upregulate PD-L1 upon cytokine stimulation, whereas mesenchymal stem cells already display a high baseline expression [36,77]. Interestingly, cholangiocytes at baseline were shown to express PD-L1 mRNA without protein expression. An IFNγ stimulation, however, was capable of inducing protein expression and altering the miRNA profile of these cells [56]. The airway epithelial cells in the lung express the PD-L1 protein already at steady state, but react to poly I:C with profound upregulation [78]. Pancreatic islet cells express low levels of PD-L1 at a steady state, which increases upon inflammation [79].

A significantly increased upregulation of PD-L1 expression, as a response to inflammatory cytokines such as IFNγ and TNF, has also been observed in keratinocytes, which have very low levels detectable at baseline [8,20,80].

In the gastrointestinal (GI) tract, PD-L1 plays an important role in mediating tolerance. High basal expression of PD-L1 was detected in epithelial cells within the gastric gland [81]. Constitutive expression of PD-L1 was found for the small intestine and colonic epithelial cells by Nakazawa et al. and suggested to play a role in the proliferation of CD4- and CD8-positive T cells [82]. In general, mesenchymal cells were the major cell type with PD-L1 surface expression in the normal human colon [83]. Located in the intestinal lamina propria are IgA plasma cells, which express high levels of PD-L1, elevated in comparison to both IgG plasma cells located in peripheral lymphoid organs as well as CD11-positive DCs, although at a lower fraction [74]. This strong presence of PD-L1 in the GI tract shows that PD-L1 plays a major role there in maintaining immune tolerance and mucosal homeostatis (reviewed in [84]). Of special interest in the context of PD-L1 are CD71^+^ erythroid cells (CECs), which not only produce erythrocytes but were shown to be highly involved in the different mechanisms of immune regulation and which can express high levels of PD-L1 [85]. These findings show that numerous cells, including cells from the hematopoietic as well as non-hematopoietic lineage, are capable of expressing PD-L1 at various levels under homeostatic or induced conditions.

## 10. Functional Consequences of PD-L1 Expression in Cell Interaction

The immune system is a complex network comprised of different cells and organs involved in different arms of protection, underlining the necessity of cell–cell communication in order to efficiently regulate and transmit signals. Signaling cascades transferred through different cell types allow the timely and spatially regulated response to different stimuli, ensuring functionality.

In allogenic T cell:DC co-cultures, a blockage of PD-L1 on immature moDCs enhanced the T-cell–originating secretion of IFNγ and IL-2, whereas this could not be observed in co-cultures with PD-L1-blocked mature moDCs, which were a priori capable of inducing strong cytokine production. However, when CD86 was blocked with antagonistic antibodies, cytokine secretion by T cells dropped. This could be reversed by additionally blocking PD-L1. DCs freshly isolated from blood behaved similarly: PD-L1 blockage on immature DCs led to an increase in T-cell-derived cytokines, but for mature DCs, CD86 blockage was not required to observe the effect [32]. When blocking PD-L1 with a single domain PD-L1 antibody on moDC, Broos et al. observed an increased capability of CD8-positive T-cell activation and enhancement of the DC-dependent TCR signaling [86].

A subpopulation of human B cells that closely resembles murine Bregs is high in PD-L1 expression and regulates the expansion and activity of follicular helper T cells, which characteristically express PD-1 [75]. Via this mechanism, Bregs are able to suppress antibody production and T-cell activation [75]. PD-L1 high-plasma cells in the small intestine lamina propria were found to be capable of inducing regulatory T cells in the periphery [74].

A functional role of PD-L1 expression in Kupffer cells emerged from experiments with cells from hepatocellular carcinoma patients. Co-culture studies revealed that PD-L1 blockage led to an increase in T-cell proliferation, as well as to their production of IFNγ, TNF, Granzyme B, and Perforin [87]. NK cells, co-incubated with tumor cells, expressed PD-L1 and were capable of directly inhibiting CD8-positive T-cell proliferation, while this inhibition could be blocked with an antagonistic PD-L1 antibody [71]. As the upregulation was found to be tumor-cell-contact dependent, and relied on the presence of certain soluble factors, it remains unclear whether this NK-cell-mediated T-cell proliferation inhibition is occurring in healthy individuals.

These results show the complex network of interactions between immune and non-immune cells. Especially in the context of an immune response, both innate and adaptive cells are taking an influence on each other, modulating their PD-L1 surface protein expression and utilizing PD-L1 itself.

## 11. Differential Cellular Location of PD-L1

PD-L1 is classically located as a transmembrane protein in the plasma membrane of cells, but during the past years, it has become evident that under certain conditions, mostly involved in the anti-cancer response, PD-L1 can be differentially located.

Exosomes—vesicles that are secreted by a cell into the extracellular space—that contain PD-L1 have been identified in vitro and in vivo in cancer research. The assessment of their influence on T cells showed that exosomal PD-L1 was capable of binding to T cells and inhibited their cytotoxic effector function, as well as NFκB activation. In vitro studies have shown the transfer of exosomal PD-L1 into a DC and macrophage precursor cell line, indicating a potential relevance in protein transfer. When assessed in vivo, the protein transfer to macrophages has been shown to be much more pronounced than to DCs, although the underlying reasons continue to remain unclear [88,89]. The exosomal presence of PD-L1 has not yet been shown in healthy tissues but has been found to be of relevance in the tumor microenvironment, potentially modulating immune surveillance and, when secreted by tumor cells, facilitating immune evasion, which also plays into their role as a potential biomarker [88,89,90].

PD-L1 has been described to enter the nucleus of different cancer cells [89], where it regulates sister chromatin cohesion [91], regulates transcription [92], and switches apoptosis to pyroptosis [89,91,93].

Exosomal and nuclear PD-L1 have yet only been shown to be of relevance in cancer cells. Further research could reveal their potential function in healthy tissue and aid in understanding whether or not the secretion of exosomal PD-L1 potentially leads to local or more widespread immunosuppression in other non-cancerous diseases such as autoimmunity.

## 12. Involvement of PD-L1 in Migration and Invasion

Apart from its immune-regulatory functions, PD-L1 appears to directly influence cell survival, migration, and invasion. The reports on this, however, all relate to tumor cells, so it is unclear to which extent this is valid for non-cancerous tissues.

Eichberger et al. showed that PD-L1 plays a role in modulation of cell mobility, more precisely cell spreading, migration, and invasion, which can in part be linked to its influence on epithelial-to-mesenchymal transition (EMT) marker expression [94]. EMT is a process involved in the migration of cells, where former resident cells express migratory markers enabling their movement. PD-L1 high-expressing cells had a lower expression of E-cadherin, which is essential for epithelial adhesion, as well as high N-cadherin and Vimentin levels, proteins directly involved in the stimulation of cell migration and invasion, or used as a marker for these functions, respectively. PD-L1 knockdown reduced cell spreading and chemotactic migration notably, while overexpression led to an increase in these functions in cells expressing moderate endogenous PD-L1 levels. Genes involved in cytoskeleton formation, which is directly linked to cell migration and cell motility, were regulated accordingly [94].

PD-L1 was also found to regulate autophagy in murine melanoma and ovarian cancer cells as the RNAi-mediated attenuation of PD-L1 expression reduced basal, as well as starvation-induced autophagy [95].

## 13. T-Cell Activating Properties of PD-L1

Upon high antigen stimulation as mimicked through anti-CD3 antibodies, PD-L1 is suppressing T-cell proliferation [7]. Nevertheless, various reports indicate a co-stimulatory effect for PD-L1 under various conditions (reviewed by [96]). Its expression on activated T cells was required for their survival, and promoted effector functions [97]. PD-L1 expression on pancreatic islet beta cells or on transplanted T cells promoted autoimmune diabetes and CD8-positive T-cell proliferation or graft-versus-host disease, respectively [98,99].

Under suboptimal CD3-stimulation conditions, PD-L1 induced T-cell proliferation [100]. CD28 co-stimulation was found to be irrelevant for the PD-L1-mediated proliferation induction. Although both CD4- and CD8-positive T cells were induced to proliferate, CD4-positive T cells were preferentially activated. Their humoral function was also enhanced, indicated by CD40L-upregulation in response to PD-L1 co-stimulation [100].

## 14. PD-L1 Involvement in Autoimmunity and Allergy

A variety of autoimmune diseases exist, and all of them are based on the breaching of the protection measurements aiming to prevent autoimmunity. The PD-1/PD-L1 pathway is of clinical and mechanistic relevance for a number of these diseases, although further research is needed for understanding the timely and spatial resolution, as well as for examining the reasons for failure. Like autoimmunity, allergy is characterized by a pathologic immune reaction against harmless antigens, which, in contrast to the former, originate from external sources such as food, pollen, medication, etc.

So far, no germline *CD274* mutations were reported, which indicates the importance of PD-L1 functionality in order for individuals to survive. In contrast, various genetic alterations in the *CD274* gene were reported somatically in cancer tissue, with gene mutation, alteration, and loss being the most prevalent occurrences, and skin cancer being the most affected cancer type [101].

In autoimmunity and allergy, PD-L1 expression is frequently elevated during the course of disease, which often counteracts disease progression or leads into a phase of remission. Further advancements in understanding the incidents tipping this tolerance mechanism could lead to the development of new therapeutic approaches, aiding in bettering the outcome or preventing clinical symptoms. An overview of the differences in PD-L1 levels between different autoimmune diseases is given in Table 2. A selection of autoimmune diseases is presented based on their general prevalence in the population and the current status of knowledge with regards to the relevance of the PD-1–PD-L1 axis.

### 14.1. PD-L1 Expression Plays a Major Role in Inflammatory Bowel Disease (IBD)

One of the important diseases worth mentioning in the context of autoimmune diseases is inflammatory bowel disease (IBD), with the two major types named Crohn’s Disease (CD) and Ulcerative Colitis (UC). Abnormal PD-L1 expression and signaling was found in both types of this gut inflammatory disease [82,118,119]. Nevertheless, contradictory findings were reported concerning the PD-L1 expression of CD and UC. Some studies demonstrated an upregulation of PD-L1 on mRNA and protein levels in UC [81,103,104], while in CD, either an upregulation or no change of PD-L1 expression in ileal disease were detected [102]. Apart from that, two studies published higher PD-L1 expression on lamina propria professional immune cells in both CD and UC [120,121]. Chulkina et al. reviewed different studies concerning PD-L1 in GI pathobiology and found numerous different effects of PD-L1 depending on the used animal model and PD-L1 source [84].

These findings show that the contribution of PD-L1 to the development and progression of IBD is still under investigation, and further studies regarding the exact role of PD-L1 signaling in this autoimmune disease are urgently needed.

### 14.2. PD-L1 Slows Progression in Multiple Sclerosis

Multiple Sclerosis (MS) is an autoimmune disease targeting the central nervous system which leads to the formation of lesions, demyelination, and a progressive loss of function. Several immune cells are involved in a network, which is not yet fully understood, with only disease-modifying therapies being available so far. Approaches aiming at the PD-1–PD-L1 axis may give rise to new, potentially specific treatments for MS, as several observations indicate a functional role of this axis in MS. Patients suffering from remitting relapsing MS express a reduced level of PD-L1 on their peripheral blood mononuclear cells (PBMCs) compared to the control, but significantly elevated levels compared to an acute MS exacerbation, linking an increase in PD-L1 levels to remission. The introduction of PD-L1 via treatment with a PD-L1-Ig fusion protein in in vitro human studies as well as in vivo MS mouse models led to disease improvement via the suppression of self-recognizing and active immune cells, promoting a tolerogenic phenotype (reviewed by [34]). A novel Treg type was found in MS, and its suppressive function via induction of Caspase-3-mediated apoptosis is positively correlated with the level of PD-L1 present [72]. During the progression of Experimental Autoimmune Encephalomyelitis (EAE), an animal model system of MS, an elevation of PD-L1 has been observed on microglial cells—the macrophage equivalent in the brain—serving potentially as a protection mechanism aiming to slow progression [105].

### 14.3. PD-L1 Expression Rises as Type I Diabetes Progresses

Type I diabetes is an autoimmune disease resulting in the destruction of the insulin-producing pancreatic β-islet cells. PD-L1 seems to counteract this because some cancer patients receiving anti-PD-L1 therapy showed the primary development of type I diabetes, which worsened through diabetic ketoacidosis [122]. In non-obese diabetic (NOD) mice, expression of PD-L1 was shown to rise significantly during progressive tissue destruction. Similarly to MS, this could be an induced counter-mechanism to limit the activity of the infiltrating self-reactive immune cells. This theory is supported by findings of β-cells expressing strong levels of PD-L1 when adjacent to infiltrating T cells, while PD-L1 blockade triggered diabetes in prediabetic NOD mice [123].

Data regarding the spatial and timely distribution showed that early phase regulation of T cells through PD-L1 was occurring rather in the pancreatic lymph nodes, and that once disease had progressed and the T cells had infiltrated the tissue, PD-L1 was rather acting locally in the affected tissue [124]. Apparently, the early expression of immunosuppressive markers is not sufficient to facilitate long-term target protection. This finding is supported by the fact that during the course of disease, PD-L1 expression increases. As soon as this measurement of autoimmunity protection fails, the system is no longer able to avert the auto-reactive T cells, resulting in an onset of the disease.

### 14.4. Loss of PD-L1 Expression on Monocytes and Myeloid Dendritic Cells during Systemic Lupus Erythematosus Flares

Systemic Lupus Erythematosus (SLE) is characterized by an attack of the immune system against self tissue and the impairment of several organs and tissues such as joints, kidney, skin, and blood vessels due to a loss of tissue tolerance by the organism. The involvement of PD-L1 in SLE has been initially shown by Mozaffarian et al. [107]. They have assessed the expression of PD-L1 in healthy individuals and SLE patients in PBMC subsets and identified the inability of immature myeloid DCs and monocytes to upregulate PD-L1 in SLE patients. This was associated with increased disease activity as PD-L1 expression was reduced during an acute flare of the disease compared to the state of remission, with respect to protein levels in general and to the number of PD-L1-positive cells. On monocytes, PD-L1 expression was two-fold higher during remission compared to healthy individuals [107]. The loss of PD-L1 during disease flares was not associated with the loss of co-stimulatory molecules, which remained at normal levels. This could be potentially detrimental, as the co-stimulation provided by CD80/CD86 is normally counteracted and modulated via the PD-1/PD-L1 axis, which appears to be removed in SLE patients.

The relevance of PD-L1 during SLE flares and the disease course in general is not conclusive so far, with variances between the different cell types. Neutrophils, in contrast to monocytes and mDCs, were found to upregulate PD-L1 upon disease flares, whereas the number of PD-L1-positive neutrophils was downregulated under the use of general immunosuppressants such as steroid treatment. The PD-L1 level on neutrophils was especially high in patients with high auto-antibody titers and high levels of inflammatory markers, suggesting neutrophil PD-L1 as a potential biomarker. Functionally, it was hypothesized that this could act as a prevention of self-inflicted tissue damage, as neutrophils expressing the immune checkpoint ligand are capable of downregulating cytotoxic autoreactive T cells [125].

Contradictory findings regarding the elevation of soluble PD-L1 have been found, with Chen et al. [108] reporting a significant increase in SLE patients compared to healthy controls, whereas Her et al. [109] were not able to show a significant difference in soluble PD-L1 levels, neither between active and remissive patients, nor compared to the control. Although the involvement of PD-L1 during SLE is most certainly of relevance, further studies involving different cell types at different disease stages would be beneficial for understanding the network and function of differentially expressed PD-L1. Limitations especially arise in human studies, since upon recognition of the disease, immunosuppressive treatment must be initiated, often preventing an assessment in the naïve SLE state.

### 14.5. PD-L1 Has a Protective Function in Rheumatoid Arthritis

Rheumatoid Arthritis (RA) is an autoimmune disease primarily affecting the joints via the formation of immune complexes, but can spread to other organs. Initial inflammation is often unspecific, but turns into a chronic disease form via the release of autoantigens due to tissue damage and the subsequent activation of autoreactive T cells and B cells [126]. The PD-1–PD-L1 pathway was found to be relevant in RA, with affected tissue expressing PD-L1 mostly in the lining and sublining layers, in contrast to control synovial tissue [110]. Both CD4- and CD8-positive T cells are expressing lower PD-1 levels compared to the control. This is in alignment with the increased activity of T cells in target tissues affected from RA, as they are numbed for PD-L1-induced suppressive signals by their lower PD-1 expression [111].

Synovial fluid myeloid DCs, i.e., DCs located in the fluid in the joints, express higher levels of PD-L1 than their PBMC mDC equivalent. This was linked to a reduction in the T-cell response when co-cultured. This effect could be countered via in vitro PD-L1 blockage, which partially restored T-cell proliferation, highlighting the protective effect of PD-L1 in RA [112]. Interestingly, the effect of this blockade was detected in vivo as well, where PD-L1 autoantibodies were identified in about 30% of RA patients. These are accompanied by an increase in T-cell proliferation, replicating the in vitro findings. As opposed to inducing proliferation in newly recruited T cells, the anti-PD-L1 autoantibodies induced apoptosis in already matured T cells when combined with a strong CD3 signal. This aberrant T-cell response of stimulating resting T cells, while simultaneously inducing apoptosis in activated T cells, could add to the worsened disease status, which is associated with the presence of PD-L1 autoantibodies [113].

### 14.6. PD-L1 Only Has a Minor Functional Role in Allergies

Allergies are harmful reactions of the body to non-pathogenic antigens, often characterized by a type II immune response. Contrary findings have been reported in the research on the impact of PD-L1. Initial studies identified that the development of allergic asthma was not influenced by PD-L1 or PD-1 as their blockage had no effect, neither during the priming phase nor the effector phase. This was determined even though PD-L1 levels were found to be elevated upon allergen challenge on some cell subtypes located in the lung, including DCs and B cells [20,115]. Opposing this, when PD-L1-deficient mice were challenged to modulate asthma, they displayed a lower level of airway hyperresponsiveness, a characteristic feature of asthma. This revealed a rather protective effect of PD-L1 deficiency [117]. Matching with the latter, in a mouse model of anaphylaxis, a reduction of specific IgE levels and mast cell degranulation was observed when PD-L1 was blocked during the sensitization phase. In contrast, a blockage during the challenge was ineffective in achieving disease modulation [116]. Other research, however, points to a potential protective function of PD-L1, as plasmacytoid DCs have been found to modulate the induction of regulatory T cells through the upregulation of PD-L1 [106].

These scarce and contradictive findings show the urgent need for a mechanistic assessment of the function in PD-L1 during allergies. However, a potential therapeutic human appliance seems challenging, because PD-L1 blockage seems to have a downmodulating effect only in the priming phase. Disease modifying therapy, such as desensitization, is aimed at the challenge phase, in which the influence of PD-L1 has not yet been pinpointed. Nevertheless, advancing research in this area could give rise to new, mechanistic findings, maybe uncovering new functions and influences of PD-L1 interaction.

In summary, it is undoubted that PD-L1 plays a crucial role in several autoimmune diseases, but due to the complex nature of the pathologic mechanisms and the multiplicity of involved cells types, the precise function of PD-L1 is not defined for most diseases. Hence, a lot of additional research is still required in this field, which certainly will pay off with respect to new therapeutic options.

## 15. The Role of PD-L1 in the Prevention of Fetal *In-Utero* Rejection

Pregnancy and fetal development pose a challenge to the immune system because, although the developing fetus is partially genetically different from the mother and therefore recognizable and rejectable by the immune system, rejection is prevented in most cases in order to allow the pregnancy to carry to term. The PD-1/PD-L1 pathway plays a vital role in this process, and PD-L1 expression increased during the progression of pregnancy. A blockage of PD-L1 in animal studies resulted in higher rates of premature terminated pregnancies and decreased litter sizes [68]. Especially in the placenta, which represents the fetal–maternal interface, where maternal blood comes into direct contact with fetal tissue, subsets of syncytiotrophoblast and cytotrophoblasts express high levels of PD-L1. The rise during pregnancy supports the hypothesis that contact with maternal blood is responsible for the increase in PD-L1 expression [68,127].

CD71^+^ erythroid cells (CECs) were found to be involved in the regulation of immune responses (reviewed in [85]). During pregnancy, CECs are enriched in the peripheral blood and in the placenta and are involved in the regulation of feto–maternal tolerance. In these pregnancy-induced CECs, high levels of PD-L1 compared to myeloid cells were observed, and it was reported that these CECs suppress T-cell responses via PD-L1 signaling to prevent an aggressive allogeneic response directed against the fetus. In contrast to this, no PD-L1 could be detected in CECs related to syngeneic pregnancy [128].

The relevance of Tregs for the maintenance of the pregnancy was discovered by Habicht et al. [129]. Tregs were previously shown to be of relevance in organ tolerance and to express high levels of both PD-1 and PD-L1. Their number increases during pregnancy, as found in animal studies and decreases postpartum. If the number of Tregs is insufficient, rates of Th1 cells, specific for paternal antigen, were found to be elevated, targeting the developing offspring and contributing to a loss of the pregnancy [130].

Mouse models have shown that in syngeneic pregnancies where the fetus is genetically identical to the mother, these protection mechanisms had no relevance, as even under PD-L1 depletion, the pregnancy was unaffected, highlighting their role in immunogenic tolerance in allogeneic pregnancies [131]. Understanding how under homeostatic conditions, an organism is able to tolerate allogeneic tissue, even allowing it to increase in size, will be beneficial into furthering transplant medicine and research.

## 16. PD-L1 Expression on Both Donor and Recipient Cells Decelerates Transplant Rejection

Allogenic transplantation is associated with the risk of rejection, as the donor is usually genetically different from the recipient. Therefore, the transplant expresses foreign antigens, which often require the recipient to receive immunosuppressive drugs. These prevent graft rejection, but render the recipient vulnerable to infections. A generation of immunogenic tolerance towards the transplanted organ would therefore be beneficial to the patient.

One of the main mechanisms involved in rejection prevention is the local induction of immunosuppression by PD-1–PD-L1 interaction. PD-L1 has been shown to be critical for the maintenance of peripheral tolerance in the context of transplantation, as blockage of the pathway led to a fastened graft rejection in CD28 and B7-1/B7-2 double deficient models. A change in balance between reactive T cells targeting the organ and tolerogenic Tregs was shown upon PD-L1 blockage, favoring pathogenic T cells and organ destruction. The expression of PD-L1 on either side, recipient cells and the donor organ, was shown to be beneficial for graft survival and tolerance, with donor PD-L1 expression limiting tissue damage and the expression on recipient APCs critical for inducing and maintaining acceptance of the organ [114].

Cardiac transplant studies have shown the upregulation of PD-L1 during the process of allogeneic rejection, whereas syngeneic transplants did not show this process. Both the transplant as well as infiltrating leukocytes expressed higher PD-L1 levels compared to controls. The co-treatment of PD-L1-Igs together with Cyclosporin A, an immunusuppressive drug, significantly enhanced allograft tolerance and survival in mice compared to mono-treatment with either substance. This is of therapeutic relevance, allowing the targeting of two arms of suppression working together in prolonging transplant acceptance [132].

Studies on professional APCs, the initiators of any specific immune response, have shown that tolerogenic DCs, which can be induced by apoptotic cells, have a meaningful influence on the induction of host and graft tolerance. Especially immature DCs are capable of prolonging tissue survival through the potential induction of Th17 cells. Liu et al. used a PD-L1-Ig fusion protein to coat immature DCs with PD-L1. These DCs had a significantly reduced capability to stimulate T-cell proliferation and decreased their Th1 cytokine secretion, while increasing Th2 cytokines. In vivo, these DCs were also capable of prolonging graft survival and inducing Tregs [133].

Understanding the involvement of PD-L1 expression in the different cell types and its exact mechanisms could aid in prolonging transplant survival, potentially giving rise to the development of certain treatments of the tissue prior to its transplantation. This could improve the life of many patients by reducing immunosuppressive drug treatment in order to prevent rejection, lowering the risk of additional infections in recipients and simultaneously prolonging transplant survival.

## 17. Conclusions

The regulation of PD-L1 expression and functionality is a complex network involving different cytokines and molecules with varying relevance of the individual modulators in different cell types. Expression of the PD-L1 protein in whole organs is very scarce and limited only to immune-privileged sites, whereas mRNA was abundantly found. PD-L1 is constitutively expressed in various cells of the hematopoietic as well as the non-hematopoietic lineage, and can additionally be upregulated to various degrees depending on the cell type and influencing modulator. Diverse functions of PD-L1 have been identified with an involvement in migration, as well as T-cell activating properties, which are exhibited under certain stimulatory conditions.

Functionally, PD-L1 is relevant for the prevention of autoimmunity under homeostatic conditions, but can become detrimental at prolonged antigen persistence, for example, during chronic infections, where it lowers the immune response due to the inhibition of T-cell function. Most advances in PD-L1 research have been made using cancer model systems, including cells and animal systems, or patients receiving anti-tumor medication. This leaves a knowledge gap when it comes to understanding the role of PD-L1 in healthy tissue and cells, and how interactions involving PD-L1 ensure homeostasis and a properly balanced function of the immune system.

Research in the area of PD-L1 expression regulation and protein function could not only be beneficial in understanding and preventing transplant rejection, but also in modulating autoimmune diseases. Knowledge of the mechanistic impact of PD-L1 in autoimmune reactions could then, in turn, help to improve cancer therapy through minimizing immune-related adverse events, which can be life-threatening and are up until now a main reason for treatment cessation.

Further studies need to address the relevance of different pathways in the various cell types, understanding the regulation of translation via microRNAs under healthy conditions, and the distribution and impact of the differentially located PD-L1 protein. There is no conclusive data available on the prevalence of germ line mutated *CD274*, which due to the potential impact of mutated protein levels or functions, could prove to be a yet-untapped resource of regulatory and functional insight into PD-L1 RNA and protein. Especially taking the high prevalence of PD-L1 mRNA in tissues into account, linked with a simultaneous absence of functional protein, the functional role of the mRNA should be examined.

Focusing on disease-related research, the impact of PD-L1, especially in terms of spatial and timely regulation, is of interest. PD-L1 has a disease-favoring role not only in cancer but also in chronic infections, where the classically protective PD-L1 expression is actually hindering the organism’s immune system from pathogen clearance. Investigating potent and, ideally, potentially locally specific PD-L1 blockage depending on the type and distribution of the pathogen, could aid in support of restoring homeostasis and the clearance of an infection.

Since in many of the presented autoimmune diseases, a changed PD-L1 expression is associated with disease severity and potential flares, advancement in evaluating this shift could aid not only in understanding the general disease mechanism, which remains elusive for many diseases, but also give hints into altering the expression through therapeutic intervention. These could prove useful to modulate or potentially ameliorate disease symptoms as early as possible, all while minimizing the side effects that are associated with the classical treatment option of systemic immunosuppression.

## Figures and Tables

**Figure 1 ijms-23-08599-f001:**
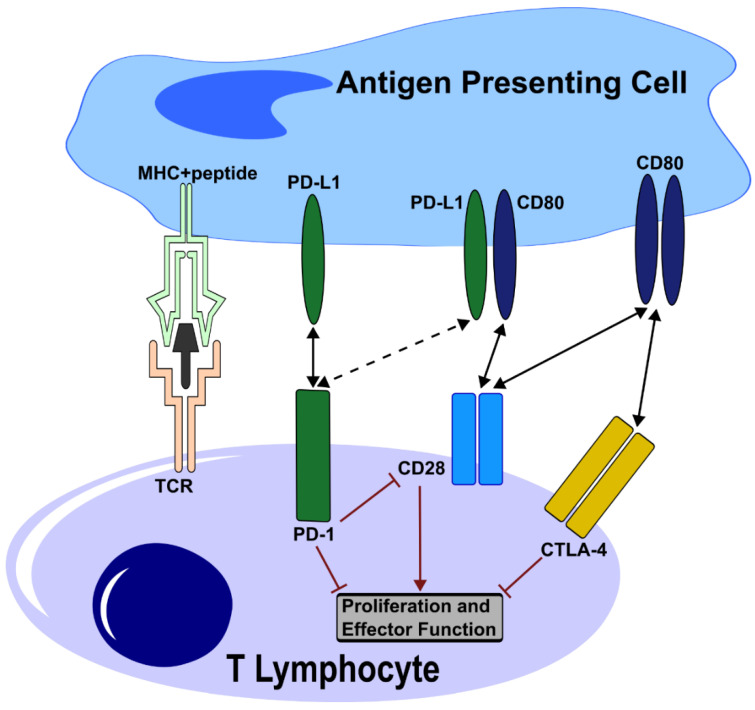
Inactivation of T-cell function through PD-L1 mediated blockage: In parallel to the activation of the TCR, PD-L1 engages PD-1, which delivers an inhibitory signal into the T cell. PD-L1 can also bind CD80 in cis, forming a heterodimer and dissolving the CD80 homodimer. This prevents the CD80–CTLA-4 interaction, while preserving CD28 binding capabilities. PD-1–PD-L1 interaction activates PD-1, leading to the dephosphorylation of CD28 and preventing T-cell co-stimulation through CD28. Black arrows depict an interaction between proteins, whereas the red hammers and arrows indicate an inhibitory and an activating effect, respectively.

**Figure 2 ijms-23-08599-f002:**
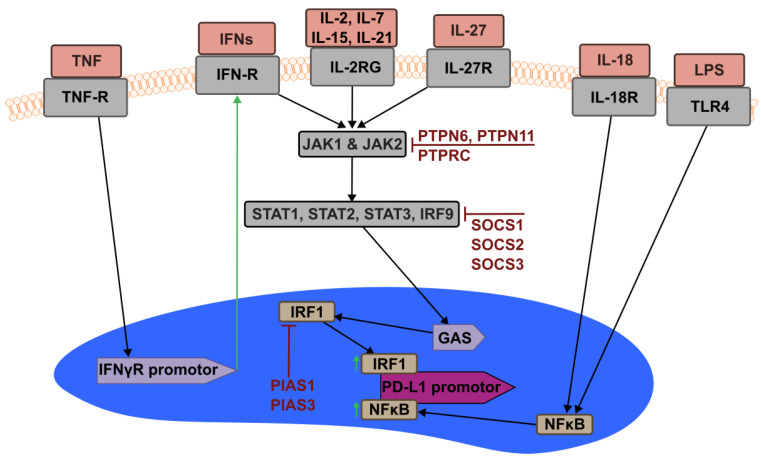
Modulation of PD-L1 Expression through Interferons and Cytokines: PD-L1 expression is initiated via different pathways, with IFN signaling being of major importance. Phosphorylation of STAT factors and subsequent signal transduction into the nucleus trigger expression of IRF1, which in turn activates the PD-L1 promotor. NFκB, activated via TLR4 signaling or IL-18, can also activate the promotor directly. TNF acts synergistically with IFN via inducing IFNγ receptor expression. PD-L1 expression is attenuated via different inhibitors able to interfere at various signaling steps. The importance of the different modulators differs depending on the cell type; please refer to the text for more detailed explanations. TNF-R: TNF-receptor; IFN-R: IFN-receptor; IL2-RG: interleukin-receptor with common gamma chain.

**Figure 3 ijms-23-08599-f003:**
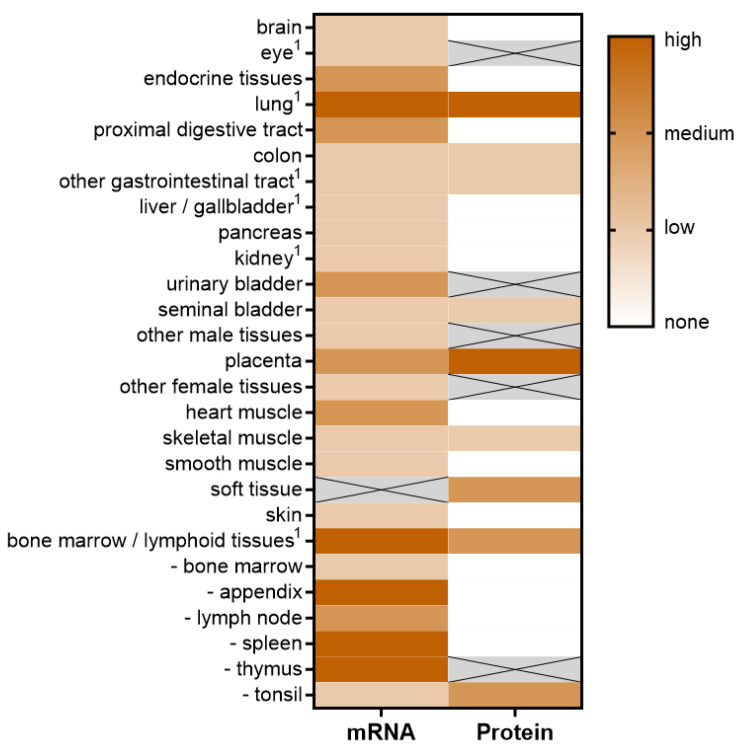
Organ PD-L1 mRNA and protein expression: RNA and protein expression are derived from the Human Protein Atlas (www.proteinatlas.org, accessed on 1 July 2022) [69,70]. The RNA expression summary is based on normalized expression (nTPM) values. Crossed-out gray cells indicate no available data. ^1^ Some authors identified a different organ expression, please refer to the main text.

**Table 1 ijms-23-08599-t001:** Overview of Cytokines and Transcription Factors Relevant for Regulation of PD-L1 Expression.

Molecule	Main Effect on PD-L1 Expression	Reference
	**Inductive signal transducers**	
JAK1	Upstream kinase: Required for PD-L1 expression; knockout leads to PD-L1 mRNA decrease	[16,23]
JAK2	Upstream kinase: Required for PD-L1 expression; knockout leads to PD-L1 mRNA decrease	[28]
TYK2	Upstream kinase: Required for PD-L1 expression	[28]
IRF1	Key factor for PD-L1 promotor function: Required for PD-L1 expression; deletion led to decreased PD-L1 expression	[28]
IRF9	Downstream transcription factor: Required for PD-L1 expression	[28]
STAT1, STAT2, STAT3	Silencing led to a minor effect on PD-L1 expression; STAT3 is required for PD-L1 expression	[28,51]
	**Suppressive signal transducers**	
PTPN11, PTPN6, PTPRC	Negative regulators of JAK1 activation	[28]
SOCS1, SOCS2, SOCS3	Inhibition of STAT factors	[28]
	**Cytokines/mediators**	
IFNα	Induction of PD-L1 expression; increases PD-L1 on DCs	[35,43]
IFNβ	Induction of PD-L1 expression	[52]
IFNγ	Major inducer of PD-L1 expression; led to higher PD-L1 mRNA expression in combination with TNF compared to IFNγ alone	[36,44,48]
IL-2	Induction of PD-L1 expression	[41]
IL-7	Induction of PD-L1 expression	
IL-10	Stimulates PD-L1 in monocytes; decreases PD-L1 expression on cancer cells	[36]
IL-15	Induction of PD-L1 expression	[41]
IL-17	Induction of PD-L1 expression; enhances PD-L1 expression in combination with IFNγ and TNF	[36,44]
IL-18	Increases PD-L1 expression on Breg	[50]
IL-21	Stimulatory effect on PD-L1 expression	[36]
IL-27	Induction of PD-L1 expression	[42,53]
TNF	Induction of PD-L1 expression; upregulation of PD-L1 in combination with IFNγ treatment	[36,44]
LPS	Upregulation of PD-L1 mRNA in human monocytes; increased expression in colorectal cancer cells	[48,49]

**Table 2 ijms-23-08599-t002:** Involvement of PD-L1 in Autoimmune Diseases and Allergies.

Disease/Dysfunction	PD-L1 Involvement	Reference
Inflammatory bowel disease (IBD)	Mainly upregulation of PD-L1 in IBD, but contradicting findings between the two major types of IBD; PD-L1 may serve as prognostic marker	[81,84,102,103,104]
Multiple sclerosis	High levels of PD-L1 slow down progression	[34,72,105]
Diabetes type 1	PD-L1 expression rises during progression	[106]
Systemic Lupus Erythematosus (SLE)	Loss of PD-L1 on immature myeloid DCs and monocytes; contradictory findings concerning soluble PD-L1	[107,108,109]
Rheumatoid arthritis	Protective function	[110,111,112,113]
Fetal *in-utero* rejection	PD-L1 increased during pregnancy; prevents fetal *in-utero* rejection	[68]
Transplant rejection	PD-L1 expression decreases during transplant rejection	[114]
Allergy	PD-L1 levels were elevated upon allergen challenge, contradictory findings: potential protective effect of PD-L1 deficiency vs. presence of PD-L1 on DCs modulate induction of Tregs	[20,106,115,116,117]

## Data Availability

Not applicable.

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
