# Peer review of "Beyond Cancer: Regulation and Function of PD-L1 in Health and Immune-Related Diseases"

_ijms, 2022, doi:10.3390/ijms23158599_

Round 1

Reviewer 1 Report

In my opinion the paper is worth publishing, since the role of PD-1/PD-L1 axis is not that fully understood in diseases other than cancerous. Yet, looking at the content of the paper and the title itself, I get the impression that the title is misleading, since the Authors concentrate on allergic and autoimmune diseases, but never inform what was the key to choose these ones over others. I believe this needs to be clarified and changed. Overall - looking at the text the text is more about PD itself than about its role in the chosen diseases - I would like the Authors to think about this. Moreover, there are some small mistakes throughout the text, please correct. 

Author Response

First of all, we thank you for reading and evaluating our manuscript. To the raised concerns we have prepared a point-by-point response:

  • “Yet, looking at the content of the paper and the title itself, I get the impression that the title is misleading, since the Authors concentrate on allergic and autoimmune diseases, but never inform what was the key to choose these ones over others. I believe this needs to be clarified and changed.”
    • Indeed, we focused on PD-L1 function and expression beyond cancer and its role in immune-related diseases, like auto-immunity and allergy, and situations where autoimmunity is a mayor threat, like transplant rejection and fetal tolerance. These are the conditions, where most is known about the involvement of PD-L1. Thus, we changed the title to: “Beyond Cancer: Regulation and Function of PD-L1 in Health and Immune-Related Diseases” and added a sentence to the text (line 601-603)
  • “Moreover, there are some small mistakes throughout the text, please correct.”
    • We have proofread the manuscript and have corrected remaining grammar and spelling errors.

Reviewer 2 Report

In their review, Beenen et al. described the regulation and function of PD-L1 in non-cancerous cells. Currently, PD-L1 is mostly investigated in the context of cancer. Therefore, the review of literature regarding the role of PD-L1 outside the cancer field is interesting.

In general, the review is well-written. Authors presents PD-L1 signaling pathways, its discovery and classification. Moreover, they discuss the regulation of PD-L1 expression and its role in the regulation of immune response in different physiological and pathological conditions.

Major points
1. Some parts of the manuscript lack references. For example first two paragraphs. It has to be corrected.

2. Figure 1  is oversimplified and is of poor quality. It has to be improved. Moreover, authors should prepare more figures that would summarize their comprehensive review.

3. I should recommend to discuss the expression of PD-L1 on immunoregulatory erythroid cells (J Immunol June 15, 2018, 200 (12) 4044-4058; DOI: https://doi.org/10.4049/jimmunol.1800113 and Pharmacology & Therapeutics, Volume 228, 2021, 107927,https://doi.org/10.1016/j.pharmthera.2021.107927) and its role in the regulation of fetomaternal tolerance (Lines 705-728)

Minor points
1. L51 - change "B7-1 and -2:" to "B7-1 and B7-2:", same for L54
2. L64-  there is some error with the reference.
3. L216 - change "JAK1, -2," to "JAK1, JAK2". Same for L217, :219.

Author Response

First of all, we thank you for reading and evaluating our manuscript. To the raised concerns we have prepared a point-by-point response:

Major:

  1. “Some parts of the manuscript lack references. For example first two paragraphs. It has to be corrected.”
    • We have added more references to back general claims.
  2. “Figure 1 is oversimplified and is of poor quality. It has to be improved. Moreover, authors should prepare more figures that would summarize their comprehensive review.”
    • We completely rebuilt Figure 1 to clarify the main interactions and effects.
    • We created a new figure (Figure 2), visualizing the influences of different cytokines and modulators on PD-L1 expression to allow for easier comprehension.
  3. “I should recommend to discuss the expression of PD-L1 on immunoregulatory erythroid cells […] and its role in the regulation of fetomaternal tolerance”
    • We have integrated the suggested references when describing PD-L1 expression of different cell types and in the section about fetomaternal tolerance as they introduce a previously not discussed cell type. (line 486-489, line758-764)

Minor:

  1. L51 - change "B7-1 and -2:" to "B7-1 and B7-2:", same for L54
    • We extended the abbreviations (B7, JAK, STAT, and the interleukins) throughout the manuscript.
  2. L64- there is some error with the reference.
    • We have fixed the reference errors.
  3. L216 - change "JAK1, -2," to "JAK1, JAK2". Same for L217, :219.
    • See above

Round 2

Reviewer 2 Report

The authors addressed all my concerns.